# Sucrose Preference Test as a Measure of Anhedonic Behavior in a Chronic Unpredictable Mild Stress Model of Depression: Outstanding Issues

**DOI:** 10.3390/brainsci12101287

**Published:** 2022-09-24

**Authors:** Dmitrii D. Markov

**Affiliations:** Institute of Molecular Genetics of National Research Centre “Kurchatov Institute”, Kurchatova Sq. 2, 123182 Moscow, Russia; molgenebio@gmail.com

**Keywords:** chronic unpredictable mild stress, anhedonia, depression, sucrose preference test

## Abstract

Despite numerous studies on the neurobiology of depression, the etiological and pathophysiological mechanisms of this disorder remain poorly understood. A large number of animal models and tests to evaluate depressive-like behavior have been developed. Chronic unpredictable mild stress (CUMS) is the most common and frequently used model of depression, and the sucrose preference test (SPT) is one of the most common tests for assessing anhedonia. However, not all laboratories can reproduce the main effects of CUMS, especially when this refers to a decrease in sucrose preference. It is also unknown how the state of anhedonia, assessed by the SPT, relates to the state of anhedonia in patients with depression. We analyzed the literature available in the PubMed database using keywords relevant to the topic of this narrative review. We hypothesize that the poor reproducibility of the CUMS model may be due to differences in sucrose consumption, which may be influenced by such factors as differences in sucrose preference concentration threshold, water and food deprivation, and differences in animals’ susceptibility to stress. We also believe that comparisons between animal and human states of anhedonia should be made with caution because there are many inconsistencies between the two, including in assessment methods. We also tried to offer some recommendations that should improve the reproducibility of the CUMS model and provide a framework for future research.

## 1. Introduction

According to the World Health Organization, more than 280 million people of all ages suffer from depression. Less than half of them receive appropriate treatment for their disorder, and, in low- and middle-income countries, 75% of people receive no treatment at all [1]. The prevalence of depression varies from 0.4% to 15.7% in different countries of the world [2]. Most modern antidepressants do not differ in their mechanism of action from their predecessors, developed almost 70 years ago. Such drugs have several disadvantages and require long-term use. Simultaneously, approximately one-third of depressed patients are considered “treatment-resistant” [3,4], and they do not experience improvement from taking antidepressants. Some researchers also question the effectiveness of antidepressants, arguing that their effects often do not differ from those of placebos [5,6,7]. It was estimated that only one in nine patients experience improvement after taking antidepressants [8]. The lack of reliable diagnostic approaches often leads to misdiagnosis [9] and, as a result, to an increase in antidepressant prescriptions without an accompanying psychiatric diagnosis [10].

The development of depression is determined by a combination of genetic and environmental risk factors. The heritability of liability to major depression is estimated at 30–50% [11,12,13]. However, neither the search for genes nor the search for single-nucleotide polymorphisms in candidate genes that would be responsible for the development of depression have yielded results [14]. Simultaneously, environmental factors—stressful situations in particular—provoke development of mental disorders, including depression [15]. Although the molecular mechanisms of the development of depression remain poorly understood, it is obvious that depression is a multifactorial disease [16]. The pathophysiology of depression can be associated with impaired levels of neurotransmitters, neurotrophins, pro-inflammatory cytokines, disturbances in clock gene machinery, impairment in the endocannabinoid system, as well as abnormal functioning of the hypothalamic–pituitary–adrenal axis (HPAA) and impaired neurogenesis in the hippocampus [17,18]. However, there is currently a lack of evidence for the leading biological theories for the onset and maintenance of depression [19].

Our society is facing a great problem, and scientists are, in the words of Eric J. Nestler, in a Catch-22 situation (a vicious cycle from which it is impossible to get out). We need animal models to better understand the pathogenesis of depression, but the animal models are difficult to develop without understanding the pathophysiology of the human disorder [20,21].

R. Porsolt, who proposed the forced swimming test in 1977, began his article with the following words: “A major problem in the search for new antidepressant drugs is the lack of animal models which both resemble depressive illness and are selectively sensitive to clinically effective antidepressant treatments” [22]. This statement has not lost its relevance at present. Despite numerous models and tests to evaluate depressive-like behavior, they all have certain shortcomings and do not completely reflect the human disorder. It is a great challenge to develop an animal model that will recapitulate all the features of human depression considering the lack of objective diagnostic criteria, the heterogeneity of depression, and the lack of an understanding of the exact etiology and pathophysiology of the disorder [23]. It is worth noting that some of the features of the depressive state that the clinician receives from the patient by subjective verbal interviews are impossible to model on animals, in particular, such feelings as sadness, worthlessness, or excessive guilt, suicidal ideation or a suicide attempt, and low self-esteem. Simultaneously, animal models are crucial and provide invaluable information in understanding the molecular mechanisms of depression, even though there is currently no ideal model of depression that fully replicates all aspects of this complex disease [24].

The CUMS model of depression is one of the best attempts to simulate the human state in animals. It is the most common and frequently used model of depression, and the sucrose preference test is one of the most common tests for assessing anhedonia in rodents. However, the difficulty of reproducing this model by different laboratories and the low reproducibility of the results impose restrictions on the use of the CUMS protocol. Completely different factors can be the source of poor reproducibility. In this review, we attempted to identify factors that might influence CUMS reproducibility and started with factors that may affect sucrose consumption (sucrose preference concentration threshold, water and food deprivation, and differences in stress susceptibility).

## 2. Methods

Original articles and reviews were searched using the PubMed database. To search for articles, the following keywords were used in various combinations: sucrose, preference, consumption, concentration threshold, chronic unpredictable mild stress, depression, anhedonia, water deprivation, food deprivation, stress-susceptible, stress-resilient. We deliberately tried to refer to primary sources, even if they were published many years ago, to avoid incorrect or ambiguous interpretations of the results. In this regard, we did not limit the search for literature to any particular period of time. Articles published more than 10 years ago were additionally analyzed in the Scopus database for newer articles, looking through all documents citing this article (original source). For each selected article, the title and abstract were analyzed for relevance to the topic of this narrative review. Since one of the goals of the review was to compare data obtained in experimental animals with data obtained in humans, we analyzed experimental articles on both animals and humans.

## 3. Chronic Unpredictable Mild Stress

There are several models of depression (learned helplessness, chronic unpredictable mild stress, chronic social defeat stress, prenatal stress, maternal deprivation, social isolation, immune stimulation, corticosterone supplementation, drug-induced withdrawal, pharmacological, and genetic models) and tests to evaluate depressive-like behavior (forced swim test, sucrose preference test, open field, social interaction, tail suspension test, light–dark box test, elevated plus maze test, and intracranial self-stimulation) [20,25,26,27,28,29,30,31].

The chronic unpredictable mild stress model is one of the most widespread and commonly used animal models of depression. For the first time, chronic stress was proposed as a model of depression in 1982 by R. Katz, who acted in an unpredictable order on animals for 3 weeks by a set of stressful factors, such as unpredictable shock, cold swim (+ 4 °C), water and food deprivation, heat stress (+ 40 °C), shaker stress, and reversal of day/night cycle, and found a decrease in the consumption of sucrose solution by animals, which was interpreted by the author as a hedonic deficit [32]. Simultaneously, the ability of antidepressants to normalize the behavioral and neuroendocrine disturbances caused by the chronic stress procedure was demonstrated [33]. However, in his paper, R. Katz used a number of quite severe (significant in strength and duration) stress factors, which, when compared with human life, did not objectively reflect the possible cause of depression.

People are constantly exposed to uncontrollable multimodal stress factors, whose nature, strength, and the probability of which they cannot predict. Strong infrequent life events and relatively weak daily hassles can both provoke the development of depression [34]. Factors that have a powerful stressful effect in our life include, for example, the death of a close family member, adultery, divorce, lawsuits, serious illness of a family member, financial problems, change in a number of arguments with spouse, retirement, sexual abuse, military operations, natural disasters, and conflicts at work [35,36,37,38]. Everyday troubles that have an annoying, worrying, disappointing effect on a person include, for example, losing things, troublesome neighbors, concerns about financial wellbeing, home maintenance, dissatisfaction with physical appearance, auto maintenance, rising prices of common goods, bad weather, nightmares, concerns about weight, overloading with work, traffic jams, and many other stressful factors that a person faces daily [39].

In 1986, Chappell P. et al., in their work, mitigated the strength of stressful factors and modified the protocol proposed by R. Katz using isolation housing, cold swim (0 °C), increased housing density, tail-pinch, double housing with unfamiliar cagemates, reversal of light/dark cycle, cold immobilization at 15 °C, food deprivation, and ether stress [40]. In 1987, P. Willner adapted the methodology of R. Katz by introducing more naturalistic stressful factors and proposed a chronic unpredictable mild stress model [41]. The following stressful factors were used by the author in an unpredictable manner in the protocol: water and food deprivation, continuous lighting, cage tilt, paired housing, soiled cage, exposure to reduced temperature (10 °C), intermittent white noise, stroboscopic lighting, exposure to an empty water bottle following a period of water deprivation, novel odors, and presence of a foreign object in the home cage. None of the above-mentioned factors, according to the authors, were necessary and sufficient to decrease sucrose intake and maintain the impairment for longer than 4 weeks [42]. Simultaneously, it was shown that antidepressants normalize the intake of sucrose solution by animals exposed to CUMS [41,43].

CUMS in rodents causes various physiological changes [44]: sleep disturbances [45], a decrease in sexual, aggressive [46], and exploratory behavior, decrease in locomotor activity and body weight, activation of HPAA with increased expression of CRH (corticotropin-releasing hormone) mRNA (messenger ribonucleic acid) in the hypothalamus, ACTH (adrenocorticotropic hormone)/corticosterone hypersecretion and adrenal hypertrophy [47,48,49], activation of the immune system [50], and changes in the levels of various neurotransmitters [51].

One of the main consequences of CUMS is anhedonia, the inability to experience pleasure, a condition that is often assessed in rodents by their preference for sucrose solution. 

Currently, according to P. Willner, there are more than 1300 publications in which CUMS was used, and, every year, the number of laboratories that use CUMS is growing. For example, in 2015, CUMS was mentioned in 230 publications and used by 180 laboratories from 30 countries [52]. However, CUMS does not show equal results among different laboratories. P. Willner himself attempted to interview 170 laboratories that had published articles using CUMS. Of the 71 respondents who answered, 75% confirmed the reliability and stable reproducibility of the model, 21% of respondents mentioned that they had difficulties with reproducibility, and 4% admitted the inability to reproduce the effects of CUMS [53]. Simultaneously, 99 respondents, for some reason, chose not to answer the question of reproducibility of the CUMS model in their laboratories.

As with any other model, CUMS has its pros and cons. The main advantages of CUMS are unpredictability and uncontrollability. The strength of the model is its naturalistic character, which best reflects the duration of exposure and the strength of stressful factors. The model’s advantages also include the long-term maintenance of induced disturbances (stability of effects), a property that allows testing drugs with a chronic administration regime and identifying therapeutic compounds with a fast mechanism of action [54,55]. Among the main disadvantages of this model are the labor-consuming nature and high sensitivity to even the smallest changes in design, which is often associated with the difficulty of reproducing the protocol by different laboratories [56]. In 1969, McKinney and Bunney proposed minimum requirements for an animal model of depression: (1) symptoms of induced depression-like state in animals must be similar to those observed in depressed humans; (2) animals must have behavioral changes that can be objectively evaluated; (3) independent observers must agree on objective criteria for drawing conclusions about a subjective state; (4) drugs that are effective in treating depression in humans must also be effective in reversing changes in animals; and paragraph 5 was reproducibility by other investigators [57]. Another disadvantage is the duration of the procedure. In the standard version adopted to date, the procedure for CUMS in rats can take up to 4 months, including an adaptation of animals to laboratory conditions (2 weeks), determination of the basal level of sucrose solution intake (7–9 weeks), CUMS (from 2 weeks until a steady decrease in sucrose solution intake appears), and chronic administration of the drug (5 weeks) [58]. The CUMS protocol is also time-consuming in mice and requires up to 9 weeks or more to complete [59]. Moreover, despite the apparent naturalism of the model, it is worth noting that animals do not encounter some of the employed stressful factors, such as continuous and stroboscopic lighting, immobilization, restraint stress, or circadian rhythm inversion, in the wild. Nevertheless, a recent meta-analysis indicated the reliability of CUMS as an animal model of depression and its strong association with anhedonia [60]. However, the results of this work may be compromised by unpublished negative results. In fact, we do not know how many research groups have tried to implement the CUMS model and failed to replicate the main effects induced by CUMS, including a decrease in sucrose preference. For example, the effectiveness of antidepressants may actually be lower than it might seem at first glance when analyzing only published data. The reason for the “ostensible effectiveness” of antidepressants probably lies in the selective reporting of clinical trial results, which could lead to publication bias [61,62]. Rosenthal R. wrote about such a trend in psychiatry as long ago as in 1979, calling such a phenomenon a file drawer problem [63]. Today, the problem of publication bias in psychiatry has not become less challenging [64]. We cannot exclude that a similar problem exists in the publication of results obtained using animal models of depression. The problem of poor reproducibility of the CUMS model remains unresolved. Strekalova et al. also recognize the need to improve the reliability and reproducibility of the CUMS model and propose to refine the methods for applying stress and evaluating behavior [65].

## 4. Subcomponents of Reward and Subtypes of Anhedonia

Criticism from a number of researchers regarding the CUMS model is directed not so much at the procedure itself, whose effectiveness on the physiological state of the animal is obvious, but at the appropriateness of using the sucrose preference test as a measure of anhedonia and depression-like behavior and the possibility of extrapolating results obtained in animals to the state of anhedonia in humans. There are some inconsistencies between rodent and human studies predominantly associated with anhedonia (Table 1). This is partly due to several issues regarding the concepts of reward and anhedonia.

It is now accepted that reward includes such subcomponents as wanting, liking, and learning [66,67,68], but, apparently, this gradation is also incomplete. Some researchers identify the following subcomponents: (1) stimulus–reward association, (2) interest/desire (wanting a reward), (3) anticipation (state of readiness for a reward), (4) motivation (initial energy expenditure to attain a reward), (5) effort (sustained energy expenditure to attain a reward), (6) hedonic response (enjoyment of reward), and (7) feedback integration (updating reward presence and values) [69,70]. However, there is still no unambiguous clarity about which of the subcomponents of the reward are actually impaired in depressed patients and animals exposed to CUMS.

The concept of anhedonia, proposed in 1886 by T. Ribot as an inability to experience pleasure, is now largely revised. Currently, anhedonia can also be divided into several subtypes, such as consummatory anhedonia (deficits in the hedonic response to rewards) and motivational anhedonia (diminished motivation to pursue rewards) [71].

Based on the nature of the rewarding stimulus in humans, at least three subtypes of anhedonia can be distinguished. Physical anhedonia is characterized by the inability to experience pleasure from eating food, touching, sex, temperature, movements, smells, and sounds. Social anhedonia is associated with the inability to experience pleasure from being with people, talking, exchanging expressions of feelings, doing things with them, competing, loving, and interacting in other ways. The third type of anhedonia is associated with intellectual pleasure, the pleasure of achievement and pleasure from art and music [72]. However, as with the reward deficits, we do not have a clear understanding of anhedonia subtypes that develop in humans with depression and in animals during CUMS.

## 5. Discrepancies in Methods to Assess Anhedonia in Rodents and Humans

Anhedonia and depressed mood are the main symptoms of depression, and the presence of one of them is a necessary condition for making a diagnosis in humans. Of these two symptoms, only the state of anhedonia seems possible not only to simulate but also to evaluate in experimental animals.

Anhedonia is currently assessed using appropriate medical scales for humans [72,73,74,75,76,77,78,79] and behavioral tests for animals (social interaction test, taste reactivity test, sucrose consumption/preference test, conditioned place preference, intracranial self-stimulation, effort-related choice behavior tasks, novel-object place conditioning, and sweet drive test) [80,81,82,83,84]. Preclinical and clinical methods of reward assessment have significant methodological differences that are related to the design of the task and the form of reward used, which leads to translational difficulties [85]. Among behavioral tests, the simplest and most convenient to use is the so-called sucrose preference test [86], which is based on the predominant consumption by animals of a sweet solution, rather than plain water, if there is a choice. There are reasons to suppose that behavioral tests are not equivalent to each other (not interchangeable) and actually evaluate different states of the animal. For example, CUMS leads to a decrease in the intake of sucrose solution by rats but does not affect the parameters of intracranial self-stimulation [87,88]. Moreover, even when using the sweet solution preference test, solutions prepared from different chemical compounds can be unequal. In particular, CUMS causes a decrease in the consumption of a 1% sucrose solution, but, at the same time, there is no change in the consumption of a 0.1% saccharin solution [89]. 

In the case of using behavioral tests to assess anhedonia in humans and animals, the type of reinforcing stimuli in these tests has a significant difference. In experimental animal studies, the so-called primary reinforcing stimuli (rewards associated with the consumption of food, fluids, sexual behavior) are used, which are instinctive and inherited. In clinical studies on patients, researchers usually employ secondary rewards acquired in the learning processes (the pleasure received from viewing photos and videos, money reward) [69]. In this regard, it is interesting to note that depressive symptoms in university females are associated with a decrease in motivation to approach secondary reinforcing stimuli (monetary and social reward) and an increase in motivation to approach primary reinforcing stimuli (food reward) [90]. Moreover, it was shown that primary (fruit juice) and secondary rewards (money) activate different brain regions [91]. This means that the presentation of different reward stimuli elicits a different neurobiological response.

The complexity of the concepts of anhedonia and reward is obvious; however, when working with experimental animals, researchers usually operate with the disturbances in the «liking» subcomponent (consummatory anhedonia), while neurobiological studies on humans focus mainly on the disturbances in the «wanting» subcomponent (motivational anhedonia). It can be assumed that, as a result of CUMS, physical, consummatory anhedonia develops, and we evaluate the hedonic response or liking. However, in this case, in our opinion, it is not correct to extrapolate a particular case of disturbance in hedonic behavior (in one of the subcomponents of reward) in animals to the human state of anhedonia, which can be more complex and multifaceted. Hayward M. interprets the two-bottle test as a measure of consummatory behavior (liking) and the progressive ratio schedule of reinforcement as a measure of appetitive behavior (wanting) [92]. Interestingly, despite a decrease in sucrose consumption during the CUMS, the same animals show no disturbances in motivation, as assessed by their performance under a progressive ratio schedule of reinforcement [93]. On the one hand, it cannot be stated that a two-bottle test evaluates only consummatory behavior, unlike, for example, a taste reactivity test, when a sweet solution is injected directly into the animal’s oral cavity [94,95]. On the other hand, the degree of motivation in a two-bottle test cannot be compared with the motivation in a progressive ratio schedule of reinforcement [96,97]. In a two-bottle test, the animal does not exert much effort to gain access to the palatable solution.

The use of different reinforcing stimuli in humans and animals makes direct comparison of results difficult. Additionally, at present, there is no clear understanding of how behavioral reactions to primary and secondary reinforcing stimuli relate to each other.

## 6. Sweet Taste Test in Humans as an Analog of Sucrose Preference Test in Rodents

The question of the appropriateness of using the sucrose preference test also arises when trying to use its analog (sweet taste test) in a clinical setting. No disturbances were observed in the preference for sucrose solution in patients suffering from depression [98,99], and no correlation was found between depressive symptoms and taste responses to sweet substances [100]; the ability to identify gustatory and olfactory stimuli remains intact [101,102]. At low concentrations of sucrose solution, the perception of depressed patients is not different from that of healthy people, but, at high concentrations, depressed patients tend to rate sucrose solutions as more pleasant but less intense [103]. It is possible that perception of the stimulus intensity might be a trait marker for depression but not wanting or liking regarding rewards [104]. Probably, there is no real change in the taste sensitivity itself, and the observed changes are associated with the interpretation of the patient’s sensation (response bias) [105]. Patients with Parkinson’s disease also deserve special attention because a significant portion of them are characterized by anhedonia. It was shown that the intensity of perception and the pleasantness of the sucrose solution in these patients remains intact [106,107], and the craving for sweets in such patients may even increase [108]. These data indicate that the state of anhedonia in humans is not obligatorily associated with changes in consumption/preference of sweet solutions, and it means that it is impossible to directly extrapolate results obtained in animals on humans. We should try to explain such discrepancies in the near future.

## 7. Is Anhedonia a Unique Symptom of Depression?

There are also no convincing answers to other questions regarding anhedonia: is anhedonia the main symptom of depression in humans? Is the inability to experience pleasure a condition typical of depression, or is it a personality trait? How specific is anhedonia to depression? Is the symptom of anhedonia homogenous or heterogeneous [109]? 

Although anhedonia is one of the main symptoms of depression in humans, it is observed in only 37% of patients [110]. Additionally, anhedonia in humans does not belong to predictive factors that suggest the onset of depression [111]. However, among treatment-resistant depressed adolescents, anhedonia represents an important negative prognostic indicator [112] and is associated with suicidal behavior [113,114,115].

It is now clear that anhedonia is not a symptom specific only to depression but is also usual for other mental disorders [116,117], including eating disorders [118,119,120,121], schizophrenia [122], substance abuse [123], Parkinson’s disease [124,125], Alzheimer’s disease [126], and even for extreme sports activity [127]. This means that the state of anhedonia is not a unique characteristic of depression and can be observed in other pathological conditions in humans. This, in turn, implies that anhedonic behavior in animals is not necessarily equal to depressive behavior.

## 8. Sucrose Preference Concentration Threshold

In the sucrose preference test, researchers usually employ a 1% solution a priori, which is not always optimal under specific conditions. The concentration of the solution should be minimal but at the same time sufficient for the preference of the solution by animals.

An analysis of sucrose preference in the concentration range from 10 mM to 100 mM among 14 different rat strains indicated a high degree of preference for the sweet solution in animals of all strains and the absence of significant differences between them [128]. However, there is evidence that, in part, consumption of sweets may depend on heredity. This is true not only for rodents [129], among which rat strains with high and low consumption of saccharin solution were bred [130], but also in humans [131,132]. Currently, three sweet-taste-liker phenotypes are distinguished in humans: the sweet-liker phenotype, the inverted U-shaped phenotype, and the sweet-disliker phenotype [133].

Rats are characterized by individual differences in the consumption of 1% sucrose solution. The consumption of sucrose solution by animals with the highest and lowest levels can differ by more than two times. Simultaneously, animals do not differ in the level of sucrose preference [134]. According to Kõiv K. et al., there are animals with high and low basal sucrose consumption. A decrease in sucrose consumption during chronic variable stress occurs only in rats with a high basal level of sucrose consumption. It means that a high basal level of sucrose intake may be predictive of higher vulnerability to stress [135]. 

The dependence of the preference for a sweet solution on age in animals is minimal. A decrease in the sucrose preference can be observed only in rats aged 69–72 weeks [136], and the taste sensitivity to sucrose solution does not change even by 2 years of age [137,138]. In humans, individuals reduce their preferred concentration of sucrose solution as they mature [139]. Younger people prefer sweeter sucrose solutions than adults [140] and have a higher preference for the sweetest orangeade [141]. According to Petty S., children had higher sucrose taste detection thresholds than adolescents, who, in turn, required higher concentrations than adults [142]. With increasing age, sensitivity to sugar increases, and the optimal preferred sugar levels decrease [143]. 

Sucrose consumption in rats has an inverted U-shape. Water consumption is minimal when there is a sucrose solution in a concentration range from 0.4 to 30%. The maximum consumption of sucrose solution is observed at a concentration of 8% [144]. The consumption of sucrose solution by rats increases proportionally with concentration, reaching a maximum at 0.25 M [145]. There is no consensus in the literature regarding the concentration of sucrose solution at which animals begin to stably distinguish it from plain water. For example, Richter C. and Campbell K., determining the taste threshold for sucrose solution for rats, found that it corresponds to 0.5% [146]. Subsequent studies indicated threshold values of 0.34% [147], 0.32% [148], 0.09% [149], and 0.08% [150].

Since the exact threshold of preference for the sucrose solution has not been established and may depend both on the state of the animals and on other factors (for example, the presence of impurities in crystalline sucrose or in the water used to dissolve it), it seems reasonable to determine the optimal concentration of the sucrose solution before the implementation of the CUMS protocol. A preliminary determination of the dependence of the preference on sucrose concentration can significantly increase the discriminatory sensitivity of the test. 

## 9. Effects of Water and Food Deprivation on the Consumption of Sucrose Solution

Criticism related to the appropriateness of using the sucrose preference test as a measure of hedonic behavior arises not only when extrapolating this test to humans but also when directly assessing the depressive-like state of animals exposed to CUMS. Initially, a number of researchers pointed to the fact that, when adjusting the sucrose consumption to the body weight of the animal, the effect of CUMS on sucrose intake disappears. Simultaneously, it was suggested that body weight largely determines the consumption of sucrose solution by animals, and that weight loss, due to food deprivation, significantly affects the consumption of sweet solution [151,152]. Forbes N. et al. argued that a reduction in sucrose solution consumption does not occur in the absence of water and food deprivation. Thus, according to the authors, water and food deprivation are necessary and sufficient conditions to reduce the consumption of sucrose solution by animals [153]. Hatcher J. et al. came to similar results and showed that a decrease in the consumption of 0.1% saccharin solution occurs only if there is food deprivation [154]. The exclusion of food deprivation from the CUMS protocol or its use at least 24 h before testing results in the absence of changes in the consumption of both sucrose solution [152,153] and saccharin solution [155]. The fact that water and food deprivation of initially stress-resilient animals is a sufficient condition to reduce their sucrose preference also indicates the effect of deprivation on the consumption of a sweet solution [156]. Simultaneously, according to P. Willner, weight loss is not a sufficient condition for reducing the consumption of sucrose solution, and, even with normalization to body weight, stressed animals still show a decrease in the consumption of a palatable solution [157]. Recent work shows that food/water deprivation does not significantly impact sucrose intake and preference in Wistar Han rats. Still, the authors recommend to exclude food/water deprivation from the protocol to avoid their stress-induced effects [158].

It is likely that food deprivation, which results in hunger and a slowdown in body weight gain, can really affect the consumption and preference of palatable solutions, but the relationship between these processes requires a separate detailed investigation. Cabanac M. proposed the term alliesthesia, conferring the dependence of the pleasantness/unpleasantness of an external stimulus for an organism on its practical usefulness, determined by the internal state of this organism [159,160]. According to the theory of ponderostat, which can be considered as a special case of alliesthesia, a decrease in body weight below a set point should correlate with an increase in the pleasure received from consuming food. The biological significance of such an increase in pleasure, according to the author, is to stimulate food intake to restore body weight to a set point [161]. Then, the likely consequence of food deprivation, accompanied by a decrease in body weight and hunger, should be an increase in the consumption of sucrose solution but not its decrease. This is confirmed by an increase in the consumption of sweet solutions [162], a decrease in the sucrose preference threshold [163], and an increase in hedonic reactions in response to the consumption of sucrose solution [164] in animals exposed to food deprivation. In humans, starvation and weight loss are known to be associated with depressive symptomatology [165] and contribute to an increase in taste sensitivity to sweet solutions [166,167]. 

Hunger caused by food deprivation affects the reward and can have a distorting effect on the interpretation of the results. In other words, the consumption of sucrose solution is determined by hunger on the one hand and the feeling of pleasure from sucrose consumption on the other. In the case of food (and/or water) deprivation, motivation associated exclusively with hunger has a great influence on the consumption of a sweet solution. The sated animal consumes a sweet solution mainly for its pleasurable or rewarding properties and a food-deprived animal mainly due to metabolic/energy needs. Food deprivation enhances the motivation to obtain food in animals, and the consumption of food under such conditions is pleasure by itself. Rats consume more sucrose pellets following food deprivation for 24 h. Under a progressive ratio schedule of reinforcement, rats have a higher break point when they were food deprived compared to food sated [168]. 

As with food deprivation, water deprivation can have a significant effect on the consumption of a sweet solution by animals. For example, it was shown that a decrease in saccharin intake and preference in rats exposed to CUMS is observed only after water deprivation [169]. This effect can be explained by the fact that, after water deprivation (in the absence of food deprivation), animals experience thirst rather than hunger, and, for this reason, they can first prefer water to a sweet solution, and, only after quenching their thirst, they begin to prefer a saccharin solution for its hedonic properties [170,171]. Additionally, it was shown that water deprivation leads to an increase in the preference threshold for sucrose solution (for animals with thirst, water tastes better) [172] and affects the rate of development of preference (animals that are not water-deprived develop preference more rapidly) [173]. 

Despite the possible shortcomings, water and food deprivation are the most commonly used stressful factors in CUMS [60]. Due to the possible effect of water and food deprivation on the consumption of sweet solutions, in our opinion, it seems reasonable to exclude these metabolic stressful factors from the CUMS protocol and replace them with other, mainly psychogenic, stressors.

## 10. No Change in Intake of Sucrose Solution during CUMS in Rodents and Increased Intake of Palatable Foods during Chronic Stress in Humans

Along with numerous data showing a decrease in sucrose consumption/preference, some authors indicate the absence of any changes in consumption [174], even in the case of manifestation of other changes that are often observed during CUMS, such as a decrease in body weight gain and adrenal hypertrophy [175,176,177]. Even a 30-day CUMS protocol was insufficient to reduce sucrose consumption and preference in rats [178]. Additionally, there is evidence of an increase in sucrose consumption by animals exposed to CUMS. When analyzing the sucrose consumption in 246 rats, an increase in consumption after 2 weeks of CUMS was found by Statham A.; however, a subsequent attempt to reproduce the effect on 146 animals did not reveal significant changes in the sucrose consumption [179]. Mice exposed to CVS (chronic variable stress) consumed more sucrose solution than non-stressed mice. Additionally, mice that were offered a sucrose solution after CVS had an increased intake compared with mice that were offered a sucrose solution before stress. The authors suggest that, after CVS, the sucrose solution is more pleasant and desirable for animals and has a greater rewarding effect [180].

It has been shown that people increase their consumption of palatable foods during chronic stress [181,182]. Depressed people consume more chocolate [183], and, in turn, the consumption of dark chocolate is associated with reduced odds of clinically relevant depressive symptoms [184]. The same trend can be observed in rodents exposed to chronic stress [185]. Administration of corticosterone to adrenalectomized animals leads to an increase in sucrose [186] and saccharin [187] consumption. Not only does chronic stress affect the consumption of palatable food but palatable food can also have a regulatory effect on hypothalamic–pituitary–adrenal axis (HPAA) functioning. The influence of sweet solutions on HPAA activity was actively studied by Dallman M. and coworkers [188], whose group has repeatedly shown that sucrose solution in high concentration has a regulatory effect on HPAA functioning [189] and that consumption of palatable food minimizes the negative consequences of HPAA activation [190]. HPAA activity is suppressed in the case of consuming sucrose and saccharin solutions, which most likely indicates a hedonic, and not metabolic, mechanism of action of sweet solutions [191,192]. This means that CUMS induces activation of the neuroendocrine system, and sucrose solution consumption at the same time dampens the stress response. This can lead to alleviation of the CUMS effects, and it is very difficult to predict the final result of these two oppositely directed processes.

It becomes clear that animals can either reduce, not change, or increase the intake of palatable solutions during CUMS. Moreover, discrepancies between laboratories are not necessarily related to the design of the experiment but rather are explained by the individual features of the animals. In particular, it was shown that some animals from one batch exposed to CUMS reduced their sucrose consumption, others did not change it, and a third portion even increased their consumption [193,194]. We should not exclude animals that increase or do not change their sucrose consumption/preference from the analysis, but we must understand the reasons for such behavior.

## 11. Differences in Stress Susceptibility (Stress-Susceptible and Stress-Resilient Animals)

Many researchers are actively discussing the individual susceptibility of an organism to stress factors. Some people successfully adapt and cope with the negative consequences of adverse effects, while others, on the contrary, become more susceptible to the development of psychopathology. A similar phenomenon has also been found in animals. It has been established that even animals of the same strain, coming from the same source, having the same sex, age, and under identical housing conditions, react differently to stressful stimuli and are divided into stress-susceptible and stress-resilient.

Numerous studies have shown that animals belonging to different groups (stress-susceptible and stress-resilient) under stressful conditions change their consumption of sweet solutions differently. It is difficult to predict how many animals belonging to one or another group will be in a particular batch. According to some researchers, 70% of animals exposed to CUMS reduce their consumption of sucrose solution, and the remaining 30% do not change their consumption [195,196,197]. Other researchers report the opposite relationship, claiming that only 35% of animals subjected to CUMS reduce their preference for sucrose solution, while 65% are stress-resilient and do not change their preference [156]. According to Ove Wiborg, during CUMS, about 20% of the animals are stress-resilient and the change in their sucrose consumption does not exceed 10%, and 40% of animals, on the contrary, are stress-susceptible and reduce their sucrose consumption by more than 30% [198,199].

The segregation of animals into stress-susceptible and stress-resilient occurs with different types of stress and is not limited to CUMS. In particular, such a distribution is observed, for example, in social defeat, where only stress-susceptible animals reduce their sucrose preference [200]. Learned helplessness can also serve as a good example of the segregation of animals into stress-susceptible and stress-resilient [201,202,203].

It should be noted that stress-resilience is an active process of adaptive maintenance of normal physiology under the action of stressful factors, which is based on the coordinated activity of all systems of the organism [204,205]. The exact mechanisms of stress-resilience are not clear, but it is known that different brain regions, various hormones, neurotransmitters, neuropeptides, and also epigenetic mechanisms are involved in adaptive changes [206,207,208,209]. Attempts are also being made to search for the genes [210], proteins [211,212,213,214], and metabolites [215] that are responsible for stress-susceptibility and stress-resilience during CUMS.

By applying the CUMS model, researchers often use the concepts of “stress-susceptibility” and “stress-resilience”. Along with this, two more terms are common in the literature: “animals with an enhanced response (high responders (HR))” and “animals with a weakened response (low responders (LR)”. In our opinion, these definitions are very close and reflect the same phenomenon: the type of response of the organism to the stress factor. Just as stress-susceptible and stress-resilient animals differ in their consumption of sweet solutions, high responders differ from low responders in depressive-like behavior. In particular, animals pre-segregated based on their locomotor activity in a novel environment (into groups of high and low responders) differ in their depressive-like behavior in the forced swim test [216], forced swim test after social defeat [217], forced swim test after chronic intraperitoneal injections [218], forced swim test after chronic stress [219], and also by sucrose preference in the model of social defeat [220].

It is obvious that animals in every particular batch are not equal and react differently to the same stressful factors. Without preliminary segregation of animals into stress-susceptible and stress-resilient groups, some effects induced by stress (anhedonia, hormone levels, etc.) can be more or less “diluted” depending on the percentage of stress-susceptible and stress-resilient animals in the group. This, in turn, can lead to the poor reproducibility of some CUMS effects.

## 12. Conclusions and Future Directions

In preparing this review, we were surprised by the lack of studies discussing the poor reproducibility of the CUMS model, studies directly comparing the state of anhedonia in animals exposed to CUMS with the state of anhedonia in humans with depression, and studies comparing various tests to assess the state of anhedonia in animals. The overwhelming majority of studies related to sucrose consumption in rodents were published more than 30 years ago and have not been confirmed by modern approaches. The CUMS model of depression is one of the best attempts to simulate the human state in animals. However, the difficulty of reproducing this model by different laboratories and the low reproducibility of the results impose restrictions on the use of the CUMS protocol. Different factors can be the source of poor reproducibility, such as suboptimal concentration of sucrose solution, water and food deprivation, and differences in stress susceptibility. There are some inconsistencies between rodent and human studies, predominantly associated with the assessment of anhedonia-like differences in reinforcing stimuli and a discrepancy between the sweet taste test and a sucrose preference test. The main issue is the appropriateness of using the sucrose preference test for the evaluation of depressive-like behavior and anhedonia. This test is very common and easy to perform. However, there are still many questions related to its employment. Is sucrose preference a measure of anhedonia? If so, which of the subcomponents of reward and which subtype of anhedonia do we evaluate? Is a change in the sucrose preference a reflection of a depression-like state or may it be associated with other processes? How do the changes in sucrose preference in animals correlate with the lack of changes in similar tests in humans? Is it possible to extrapolate the results obtained in animals to the state of anhedonia in humans? Why does a change in sucrose preference in existing cases not correlate with a change in the threshold of intracranial self-stimulation? We still have to answer not only these but also many other questions regarding the neurobiology of depressive disorder and experimental modeling of depressive state in animals.

We can offer the following main recommendations regarding the use of the sucrose preference test aimed at improving the reproducibility of the CUMS model:

(1)Determine the minimal (or optimal) concentration of the sucrose solution before the CUMS.(2)Determine the basal level of sucrose consumption/preference before the CUMS to identify animals with low, high and/or unstable sucrose consumption.(3)To minimize the influence of metabolic factors on animals’ sucrose preference, it is advisable to exclude water and food deprivation from the protocol or evaluate sucrose preference not earlier than 24 h after deprivation.(4)Before starting the experiment, it is advisable to segregate animals into groups with high and low responses to stressors (stress-susceptible and stress-resilient animals).

A deep study of the following questions seems to be very interesting:

(1)Analysis of sucrose consumption (sucrose preference concentration threshold, dependence of sucrose consumption on age, sex, strain, time of day, physical and emotional state, etc.)(2)Comparison of different tests for assessing anhedonia in animals with each other.(3)Revealing reward subcomponents that are actually impaired in depressed patients and animals exposed to CUMS(4)Characterization of anhedonia subtypes, which develop in depressed humans and animals exposed to CUMS(5)Studying the regulatory effects and mechanisms of action of palatable food on HPAA functioning during CUMS(6)Studying the relationship between hunger (food deprivation) and sucrose solution consumption by rodents.

## Figures and Tables

**Table 1 brainsci-12-01287-t001:** Major differences between humans and animals related to anhedonia.

Humans	Animals
**The evaluated subtype of anhedonia**	
motivational anhedonia	consummatory anhedonia
**Assessed subcomponents of reward**	
wanting	liking
**Methods for assessing anhedonia**	
mostly medical scales	behavioral tests
**The type of reinforcing stimuli to assess anhedonia**	
secondary reinforcing stimuli (money reward, etc.)	primary reinforcing stimuli (food, fluid)
**Preference of sucrose solution in depressive state**	
lack of changes in depressive patients	decreased preference in animals with depressive-like behavior
**Consumption/preference for sweet food/fluid during chronic stress**	
an increase in consumption of palatable food	a decrease in sucrose consumption/preference

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
