# Peer review of "Sucrose Preference Test as a Measure of Anhedonic Behavior in a Chronic Unpredictable Mild Stress Model of Depression: Outstanding Issues"

_brainsci, 2022, doi:10.3390/brainsci12101287_

Round 1
Reviewer 1 Report
The author reviewed studies in a chronic unpredictable mild stress model (CUMS) of depression. The CUMS model of depression is one of the best attempts to simulate human state in animals. However, the difficulty of reproducing this model by different laboratories and the low reproducibility of the results impose questions the use of the CUMS protocol for depression studies. The author pointed out some outstanding issues related to CUMS. They pointed out some inconsistencies between rodent and human studies predominantly associated with the assessment of anhedonia, and stated that the main issue is the appropriateness of using the sucrose preference test for the evaluation of depressive-like behavior and anhedonia. I find this review is very interesting and this will add important insight into animal-human studies of depression. I only have a few minor comments which I hope will help improve this manuscript.
1. “At the same time, 36% of patients are treatment-resistant (Fava and Davidson, 1996) and they do not experience improvement from taking antidepressants.”
-This is a relatively old reference to be cited.
2. Please describe “CUMS” for the first use in the main text as well as in the abstract.
3. Please describe “CRH”, “mRNA”, “HPAA”, and “ACTH” for the first use.
4. “In 1969, McKinney and Bunney proposed minimum requirements for an animal model of depression”
-I would suggest summarizing this minimum requirement in the main text or in the table. What is the difference between McKinney and Bunney (1969)’s requirements and the author's recommendations on page 11?
5. Antoniuk et al., 2019 also mentioned the heterogeneity CUMS studies. I would suggest mentioning their work here.
6. Please check font sizes and citation styles.
Author Response
Dear reviewer, thank you very much for your attention to our review. Please see our answers in the attached file.

Reviewer 2 Report
This is an interesting review about the sucrose preference test for the assessment of anhedonic behavior in models of depression. The topic is of interest for the readers, and interesting for the journal. However, several minor changes should be made before considering it for publication.
Abstract.
1- The abstract is not structured into Background, Methods, Results and Conclusions. I recommend to divide it according to these subsections. How did the author the review? A brief description of the method should be described.
2- CUMS has been abbreviation without previous mention of "Chronic unpredictable mild test".
Introduction
1- The introduction section is really short. In a first part, the authors are introducing some words about depression and its epidemiological findings. I recommend to add some references about the neurobiology of depression and main biological underpinnings of antidepressant response. This would be a good introduction for the subsequent review of the tests.
2- At the end of the introduction section, I recommend to add some words about the main "Aims and objectives" of the study.
Methods
1- There is no methods section. If the authors carried out a narrative review, they should describe it. How did the author search studies on this topic?
Please, add the screening and selection process description, search terms used, etc. There is a checklist called SANRA that could help the author.
Results
1- The author starts the review with the Chronic Unpredictable Mild Stress model of depression. I recommend to add other tests used as models of depression before considering the CUMS.
2- Description of several models of depression would be of interest to introduce section 3. "Subcomponents of reward and subtypes of anhedonia".
3- Discrepancies in methods to assess anhedonia in rodents and humans should be described at the beginning of the paper, because the selection of the CUMS should be based on these informations.
4- In the results section, the authors are describing clinical results (6. Is anhedonia a unique symptom of depression) and other basic informations (7. Sucrose preference concentration threshold). I recommend to separate them into two main sections, and then build subsections.
A limitations and strengths section is needed.
The numbering and references style should be revised.
Author Response

(The authors gave the same response as above.)

Reviewer 3 Report
In the manuscript “Sucrose preference test as a measure of anhedonic behavior in a chronic unpredictable mild stress model of depression. Out-standing issues”, Dmitrii D. Markov discussed the main discrepancies between rodent and human studies in assessing anhedonia.
In my opinion, the manuscript is original and deals with an engaging thematic. Therefore, the paper reaches an interesting level of novelty, characterizing articles published in Brain Sciences, and it can be considered for publication after addressing the points:
1) The English form of this paper needs to be revised, thus permitting the reading of the paper to be more fluent.
2) Where possible Authors should update references.
3) In order to enrich the paper, I suggest Authors to include summary tables at the end of each paragraph.
Author Response

(The authors gave the same response as above.)
